**Subject Area:**
biochemistry/cellular biology/genetics

glycosylphosphatidylinositol, post-translational modification, biosynthetic pathway, GPI deficiency, protein shedding

**Author for correspondence:**
Taroh Kinoshita
e-mail: tkinoshi@biken.osaka-u.ac.jp

# Biosynthesis and biology of mammalian GPI-anchored proteins

Taroh Kinoshita

Yabumoto Department of Intractable Disease Research, Research Institute for Microbial Diseases, Osaka University, 3-1 Yamadaoka, Suita, Osaka, Japan

(iD) TK, 0000-0001-7166-7257

At least 150 human proteins are glycosylphosphatidylinositol-anchored proteins (GPI-APs). The protein moiety of GPI-APs lacking transmembrane domains is anchored to the plasma membrane with GPI covalently attached to the C-terminus. The GPI consists of the conserved core glycan, phosphatidylinositol and glycan side chains. The entire GPI-AP is anchored to the outer leaflet of the lipid bilayer by insertion of fatty chains of phosphatidylinositol. Because of GPI-dependent membrane anchoring, GPI-APs have some unique characteristics. The most prominent feature of GPI-APs is their association with membrane microdomains or membrane rafts. In the polarized cells such as epithelial cells, many GPI-APs are exclusively expressed in the apical surfaces, whereas some GPI-APs are preferentially expressed in the basolateral surfaces. Several GPI-APs act as transcytotic transporters carrying their ligands from one compartment to another. Some GPI-APs are shed from the membrane after cleavage within the GPI by a GPI-specific phospholipase or a glycosidase. In this review, I will summarize the current understanding of GPI-AP biosynthesis in mammalian cells and discuss examples of GPI-dependent functions of mammalian GPI-APs.

## 1. Introduction

At least 150 human proteins are glycosylphosphatidylinositol-anchored proteins (GPI-APs) [1]. GPI-APs are integral membrane proteins present on the cell surface. The protein moiety of GPI-APs is basically hydrophilic lacking transmembrane domains, and is anchored to the plasma membrane (PM) with GPI moiety covalently attached to the C-terminus (figure 1). Sizes of the proteins range widely from only 12 amino acids (CD52 [2]) to greater than 200 kDa (alpha-tectorin [3]). Their functions also vary, including hydrolytic enzymes, receptors, adhesion molecules, protease inhibitors, complement regulators and prions (table 1).

The GPI moiety of GPI-APs consists of the conserved core glycan, phosphatidylinositol (PI) and glycan side chains. The structure of the core glycan is EtNP-6Manα2-Manα6-(EtNP)2Manα4-GlNα6-myoIno-P-lipid (EtNP, ethanolamine phosphate; Man, mannose; GlcN, glucosamine; Ino, inositol) [4] (figure 1). The GPI is linked to the C-terminus via an amide bond generated between the C-terminal carboxyl group and an amino group of the terminal EtNP [4]. The entire GPI-AP is anchored to the outer leaflet of the lipid bilayer by insertion of hydrocarbon chains of PI (figure 1).

GPI-anchoring is a post-translational modification. The core GPI is assembled on the endoplasmic reticulum (ER) membrane and is transferred en bloc to the precursor proteins immediately after their ER translocation (figure 2). The nascent GPI-APs, in which GPI structure is immature, undergo several remodelling steps in the ER and the Golgi apparatus (figure 3). For many GPI-APs, a side chain glycan is added in the Golgi. Finally, GPI-APs are transported to the PM (figure 3).

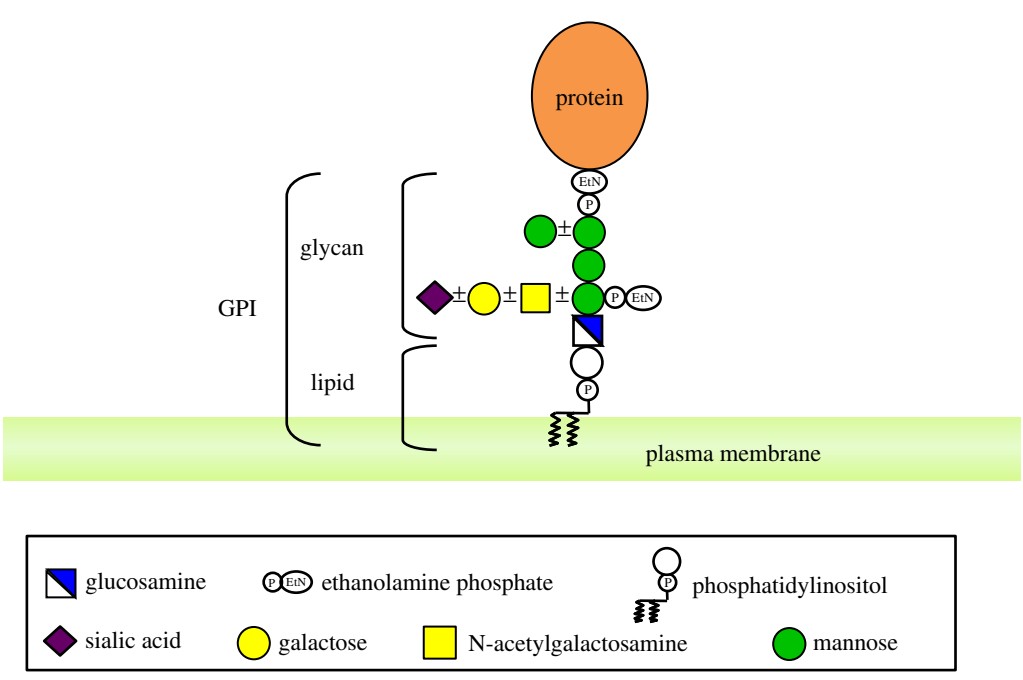

**Figure 1.** Mammalian GPI-APs. The conserved core glycan of mammalian GPI, which consists of EtNP attached to the protein, three Mans, EtNP attached to Man1, and GlcN, is linked to the lipid moiety, which is PI. In some GPI-APs, the core glycan is modified by Man4 and/or GalNAc side chains. The GalNAc side chain can be elongated by Gal and Sia. The entire GPI-AP is anchored to the outer leaflet of PM only by hydrocarbon chains of PI.

Because of GPI-dependent membrane anchoring, GPI-APs have some unique characteristics. A prominent feature of GPI-APs is their association with membrane microdomains or membrane rafts [5,6]. The membrane microdomains are dynamic domains of 20–100 nm width [7]. Sphingolipids and cholesterol are enriched in the outer leaflet of the lipid bilayer in the microdomains. GPI-APs are thought to associate with glycosphingolipids and cholesterols via lipid–lipid and glycan–glycan interactions. For lipid–lipid interactions, saturated lipid chains of both sphingolipids and GPI anchors are critical [8,9]. Within the membrane microdomains, GPI-APs form dimers with a half-life of 100 ms [10]. For GPI-AP homodimer formation, protein–protein interactions are critical [10,11]. GPI-APs-containing microdomains are regulated by cortical actin [12]. For the regulation by cortical actin, transbilayer interaction between lipid moiety of GPI-APs in the outer leaflet and phosphatidylserine in the inner leaflet is critical [13]. Src family tyrosine kinases [14] and other proteins such as phospholipase Cγ (PLCγ) are associated with membrane microdomains in the inner leaflet. Upon ligation, GPI-APs make bigger clusters that lead to activation of Src-tyrosine kinases and PLCγ [15,16].

In the polarized cells such as epithelial cells, many GPI-APs are exclusively expressed in the apical surfaces [17] whereas some GPI-APs are preferentially expressed in the basolateral surfaces [18] (see the reviews for mechanisms of GPI-APs sorting [19–22]). Several GPI-APs act as transcytotic transporters carrying their ligands from one compartment to another [23–25]. Some GPI-APs are shed from the membrane after cleavage within the GPI by a GPI-specific phospholipase or a glycosidase [26–29].

In this review, I will summarize the current understanding of GPI-AP biosynthesis in mammalian cells and discuss examples of GPI-dependent functions of mammalian GPI-APs.

# 2. Biogenesis of GPI-APs: post-translational modification of proteins with the GPI mediated by GPI transamidase

## 2.1. Translocation of the precursor proteins into the ER

The precursor proteins of GPI-APs, preproproteins, have an N-terminal signal peptide for ER translocation and a C-terminal signal peptide for GPI attachment (figure 4) [30]. The N-terminal signal peptide consisting of about 20 hydrophobic amino acids is similar to those of other secretory proteins [30]. The C-terminal GPI attachment signal peptide spans 20–30 amino acids starting from the ω+1 amino acid (the amino acid to which GPI is attached is termed the ω site amino acid [30]). The GPI attachment signal peptide consists of a stretch of about 10 hydrophilic amino acids and a stretch of about 20 hydrophobic amino acids. Amino acids at the ω site are always small ones, including Ser, Asn, Asp, Ala, Gly, Cys and Thr, and the ω+2 amino acid is also small [31–33]. In some GPI-APs, more than two ω sites were identified [33]. The GPI attachment signal peptide can direct GPI attachment to non-GPI-APs if it is fused to the C-terminus (see the review for further discussion [34]). Upon translocation of a preproprotein into the ER lumen, the N-terminal signal peptide is removed from the precursors generating proproteins (figure 4). The C-terminal signal peptide is recognized by the GPI transamidase, which cleaves and replaces it with a preassembled GPI by transamidation, generating nascent GPI-APs [30,35,36].

Mechanisms of ER translocation of two GPI-APs, the prion protein and CD59, are different. The prion requires Sec62 and Sec63 whereas CD59 does not [37,38], suggesting that the prion translocates in a signal recognition particle (SRP)-independent manner whereas CD59 translocates in an SRP-dependent manner [37]. The dependency of the

royalsocietypublishing.org/journal/rsob Open Biol. 10: 190290

**Table 1.** Examples of mammalian GPI-APs.

| GPI-AP | functional category | roles/characteristics |
| --- | --- | --- |
| alpha-tectorin | unknown | hearing/largest GPI-AP |
| CD52 | unknown | T-cell CAMPATH-1 antigen/smallest GPI-AP |
| CD55 (DAF) | complement inhibitor | self-damage protection |
| CD59 | complement inhibitor | self-damage protection |
| CRIPTO-1 | co-receptor | morphogenesis/shedding |
| dipeptidase 1 | enzyme | hydrolysis of various dipeptides |
| folate receptor 1 | receptor | folate uptake/transcytosis |
| GP2 | receptor | mucosal immunity/transcytosis |
| GPIHBP1 | receptor | transcytosis of lipoprotein lipase |
| LY6 K | sperm protein | fertilization |
| prion | unknown | prion disease agent |
| RECK | protease inhibitor | neurogenesis/shedding |
| TEX101 | sperm protein | fertilization/shedding |
| Thy1 | unknown | neuronal protein |
| uPAR (CD87) | receptor | binding of urokinase plasminogen activator/shedding |
| contactins | adhesion molecules | cell adhesion |
| tissue non-specific alkaline phosphatase | enzyme | uptake of vitamin B6 and other functions |

prion upon Sec62 and Sec63 is determined by its N-terminal signal sequence [37]. Studies on ER translocation of *Saccharomyces cerevisiae* GPI-APs showed that their ER translocation is mainly Sec62- and Sec63-dependent [39]. The study further demonstrated that they are translocated into the ER through a post-translational mechanism, to which the C-terminal GPI attachment signal peptide also contributes [39]. For GPI-APs' precursor bearing a strongly hydrophobic C-terminal peptide, components of the GET pathway, which have a role in ER incorporation of tail-anchored proteins [40], are involved. SRP-dependent co-translational ER translocation has a minor role relative to a post-translational mechanism in yeast [39]. Whether ER translocation of mammalian GPI-APs, other than the prion protein, is mediated by a post- or co-translational mechanism is yet to be characterized.

## 2.2. GPI transamidase

GPI transamidase is an ER-resident enzyme complex that mediates GPI-anchor attachment to proteins [41,42]. GPI transamidase cleaves the GPI attachment signal peptide between the $\omega$ and $\omega+1$ amino acids, generating a substrate–enzyme intermediate linked by a thioester bond between the $\omega$ amino acid carboxyl group and a catalytic cysteine side chain of the enzyme. The thioester bond is attacked by an amino group of the terminal EtN of GPI, completing a transfer of GPI by transamidation [35].

GPI transamidase consists of five subunits, PIGK (initially termed GPI8) [43], GPAA1 (initially termed GAA1) [44], PIGS [45], PIGT [45] and PIGU [46] (table 2). PIGK, a single transmembrane protein, is a cysteine protease that cleaves the C-terminal peptide and makes a carbonyl intermediate [43]. GPAA1, a multiple transmembrane protein having sequence homology to an M28 family peptide-forming enzyme, seems to catalyse the formation of an amide bond between the $\omega$ amino acid and GPI's EtN [47]. PIGT, a single transmembrane protein, associates with PIGK via a disulfide bond, thereby playing a role in complex formation [48]. The roles of PIGS and PIGU, both being multiple transmembrane proteins, have remained unknown; however, both are essential for the activity of GPI transamidase [45].

## 2.3. Biosynthetic assembly of GPI precursors

Biosynthesis of GPI is a stepwise sequence of 11 reactions (figure 2 and table 2). The pathway is initiated on the cytoplasmic side of the ER by GPI N-acetylglucosaminyl transferase (GPI-GnT), which catalyses the transfer of N-acetylglucosamine (GlcNAc) from uridine diphosphate (UDP)-GlcNAc to the 6-position of inositol to generate GlcNAc-PI. GPI-GnT is the most complex monoglycosyltransferase, consisting of seven subunits, PIGA [49], PIGC [50], PIGH [51], PIGQ (initially termed GPI1) [52], PIGP [53], PIGY [54] and DPM2 [53], of which PIGA is a catalytic subunit. PIGC, PIGH, PIGP and PIGY are essential for the activity of GPI-GnT although their specific functions are not clear [50,51,53,54]. PIGQ stabilizes a core complex of PIGA, PIGC and PIGH [55], whereas DPM2 enhances the GPI-GnT activity by three times [53]. GlcNAc-PI is de-N-acetylated by PIGL, an ER-resident GPI deacetylase, that generates the second intermediate glucosaminyl (GlcN)-PI [56,57]. GlcN-PI flips into the luminal side, the mechanism of flipping being unknown. The 2-position of the inositol ring of GlcN-PI is acylated by acyltransferase PIGW, a multiple transmembrane protein, to generate GlcN-(acyl)PI [58]. The functionally important amino acids in PIGW and its yeast orthologue Gwt1p reside on the luminal side, suggesting that palmitoyl-CoA is available on the luminal side [58,59].

PI, GlcNAc-PI and GlcN-PI in mammalian cells contain diacylglycerol, whereas GlcN-(acyl)PI contains 1-alkyl-2-acylglycerol as a major form and diacylglycerol as a minor form, suggesting that diacyl to 1-alkyl-2-acyl lipid remodelling occurs at the stage of GlcN-(acyl)PI [60]. Fatty acyl chain analysis demonstrated that the chain composition of diacyl glycerol of GlcN-(acyl)PI is clearly different from that of PI, GlcNAc-PI and GlcN-PI [61]. In the latter, 1-stearoyl-2-arachidonoyl (38:4) PI is by far the most abundant form. By contrast, less than 30% of GlcN-(acyl)PI had a chain

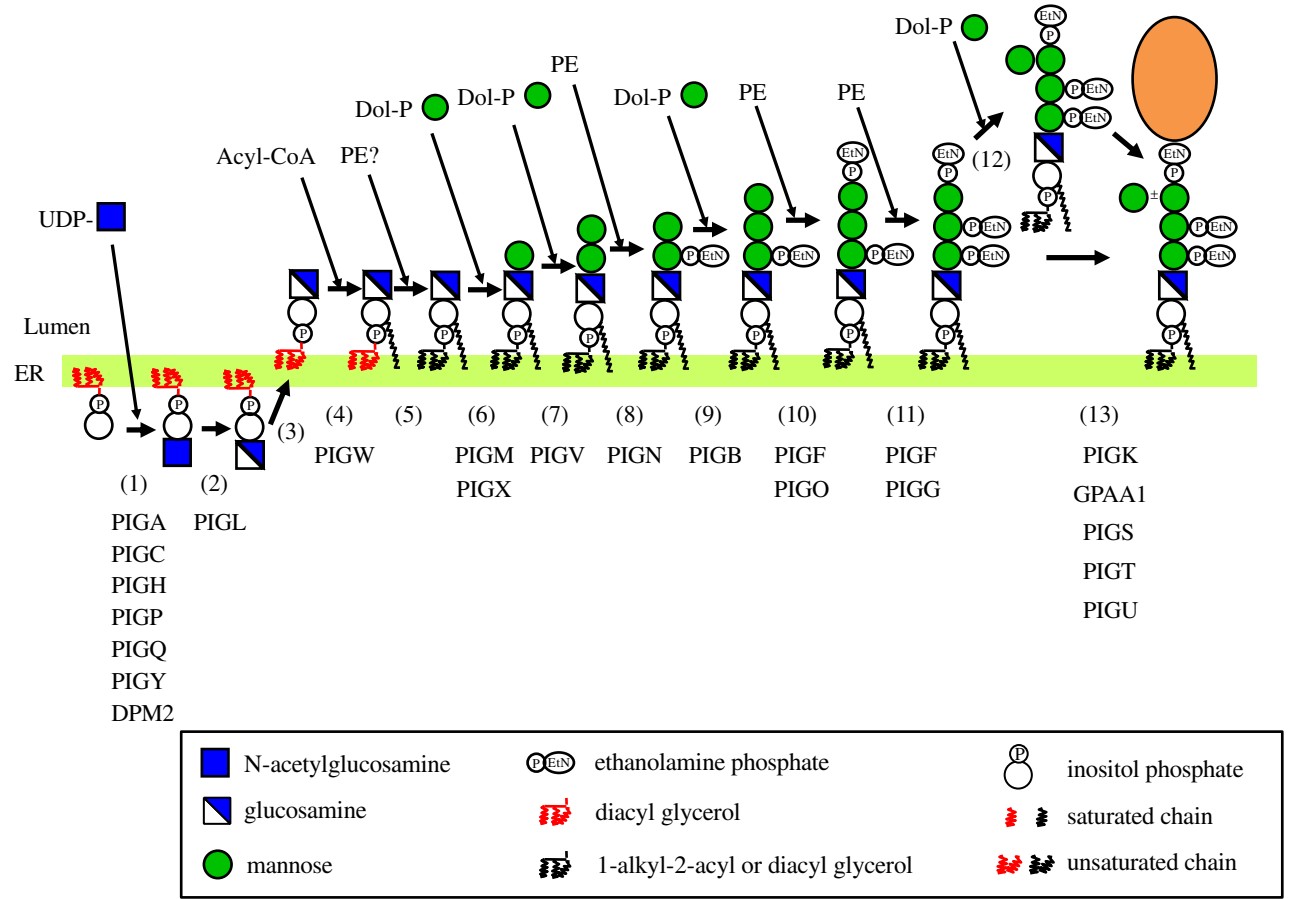

**Figure 2.** Biosynthesis of mammalian GPI in the ER. The complete GPI precursor competent for attachment to proteins is synthesized from PI by stepwise reactions (1)–(11). The Man4 side chain is attached in the ER to some GPI (step (12)). The preassembled GPI is en bloc transferred to proteins (step (13)). Genes involved in these reaction steps are shown below step numbers.

composition of 38 : 4 whereas the rest of GlcN-(acyl)PI had chain compositions of 36 : 4, 38 : 5 and 40 : 5. Therefore, it is highly likely that the lipid remodelling is diacyl to diradyl remodelling, in which not only 1-alkyl-2-acyl PI but also diacyl GlcN-(acyl)PI are the remodelled products [61]. For generation of the 1-alkyl-2-acyl form, 1-alkyl phospholipid synthetic pathway in the peroxisome is required [61]. Although the exact enzymatic reaction of the lipid remodelling is unclear, it is conceivable that either diacyl glycerol or diacyl-phosphatidyl moiety is replaced by a corresponding diradyl structure. A donor of diradyl structure is also unknown; however, the fatty chain composition of GlcN-(acyl)PI is somewhat similar to that of diradyl phosphatidylethanolamines (PE), suggesting their possible contribution [61].

Two mannoses (Man1 and Man2) are sequentially transferred from dolichol-phosphate-mannose (Dol-P-Man) to GlcN-(acyl)PI to generate Manα6Manα4GlcN-(acyl)PI. PIGM and PIGV are GPI-mannosyltransferase (MT) I and II, respectively, that catalyse these reactions [62,63]. PIGM functions in association with PIGX, which stabilizes PIGM [64]. GPI-ethanolaminetransferase I (ETI), PIGN, transfers EtNP from PE to the 2-position of the first, α4 linked Man generating Manα6(EtNP)2Manα4GlcN-(acyl)PI [65]. The third Man (Man3) is then transferred from Dol-P-Man by PIGB, which is GPI-MTIII, to generate Manα2 Manα6(EtNP)2Manα4GlcN-(acyl)PI [66]. Two EtNPs are sequentially transferred to the 6-positions of Man3 and then Man2 by PIGO [46] and PIGG (initially termed GPI7) [67], catalytic subunits of GPI-ETII and GPI-ETIII. Both PIGO and PIGG are stabilized by association with PIGF [46,67,68]. The resulting EtNP6Manα2(EtNP)6Manα6(EtNP)2Manα4GlcN-(acyl)PI is a mature GPI precursor competent for attachment to proteins.

The fourth Man (Man4) may further be transferred from Dol-P-Man to the 2-position of Man3 in the mature GPI precursor as a side chain to generate Manα2(EtNP)6Manα2(EtNP)6Manα6(EtNP)2Manα4GlcN-(acyl)PI, which is also competent for attachment to proteins. Transfer of Man4 is mediated by PIGZ (initially termed SMP3), GPI-MTIV [69]. All four GPI-MTs (PIGM, PIGV, PIGB and PIGZ) are multiple transmembrane proteins bearing catalytic sites within the luminal regions [63]. Three GPI-ETs (PIGN, PIGO and PIGG) are also multiple transmembrane proteins bearing catalytic sites within the luminal regions [67]. The bioinformatic study identified a common GPI recognition region in transmembrane domains of PIGW, PIGM, PIGV, PIGB, PIGZ, PIGN, PIGO and PIGG, and suggested a gene family comprising them [70].

# 3. GPI-AP maturation pathway and ER-to-Golgi transport of GPI-APs

## 3.1. GPI remodelling in the ER and ER-to-Golgi transport

The GPI moiety in the nascent GPI-APs is immature in its structure and undergoes remodelling reactions to become mature during transport to the PM (figure 3 and table 2). Soon after generation of the nascent GPI-APs, the acyl

**Figure 3.** Maturation of mammalian GPI-APs during ER–PM transport. Nascent GPI-APs generated by the transfer of GPIs to proteins (step 12) undergo two reactions, inositol-deacylation (step 14) and removal of the EtNP side chain from Man2 (step 15) in the ER. The ER–Golgi transport of GPI-APs is mediated by COPII-coated vesicles (step 16). In the Golgi apparatus, GPI-APs undergo fatty acid remodelling (steps 17 and 18). Some GPI-APs is modified by the GalNAc side chain (steps 19–21). The mature GPI-APs are transported to the PM where they are associated with raft microdomains. Genes involved in these reaction steps are shown below step numbers.

chain linked to the inositol ring is usually removed by GPI-deacylase PGAP1 (for Post GPI Attachment to Proteins 1), an ER-resident multiple transmembrane protein [71]. In some cases, the acyl chain remains in the mature GPI-APs [72]. For example, GPI-APs in erythrocytes maintain the inositol-linked acyl chain [73,74]. GPI-APs that have (acyl)PI, which has three hydrocarbon chains, associate with the membrane more stably than those that have PI, which has two hydrocarbon chains. The former structure might be useful for maintaining levels of GPI-APs during the very long life of erythrocytes by reducing spontaneous release from the PM.

While still in the ER, EtNP linked to Man2 is removed by a phosphodiesterase PGAP5/MPPE1 [75]. PGAP5, a membrane protein bearing two transmembrane domains, exists near the ER exit sites in the ER and in the ERGIC [75]. After the remodelling of the GPI by PGAP1 and PGAP5, GPI-APs are recruited into the COPII-coated transport vesicles that are directed towards the Golgi apparatus [75,76]. A complex of p24α2, p24β1, p24γ2 and p24δ1 of the p24 family of proteins acts as a cargo receptor for recruitment of GPI-APs into the transport vesicles at the ER exit sites [76–78]. The remodelling reactions by PGAP1 and PGAP5 are important to generate the proper structure of the GPI for association with the cargo receptor [76]. A defect in PGAP1 or PGAP5 results in slowed transport of GPI-APs from the ER to the Golgi. After arrival at the Golgi, GPI-APs are released from the cargo receptors under acidic conditions.

## 3.2. Fatty acid remodelling in the Golgi

As described in the previous section, the PI moiety of GPI-APs is initially derived from cellular PI and is subjected to lipid remodelling at the stage of GlcN-(acyl)PI, in which the original diacyl PI moiety is remodelled to a mixture of diacyl and 1-alkyl-2-acyl PIs, with the latter being the major form [60,61] (step 5 in figure 2). The fatty chain composition of the remodelled diradyl PI should correspond to that in the donor lipid, possibly PE, used in the lipid remodelling reaction [61]. The alkyl chain and the 1-acyl chain in the diradyl PIs are mostly saturated C18 or C16 alkyl and C18 acyl chains, whereas the 2-acyl chains are unsaturated fatty acids of the C16–C24 chains [61]. In the Golgi apparatus, the unsaturated 2-acyl chain is replaced with a saturated fatty acid, mostly stearic acid by fatty acid remodelling [79] (steps 17 and 18 in figure 3). In the first step of the fatty acid remodelling, the unsaturated 2-acyl chain is removed by a GPI-specific phospholipase $A_2$ PGAP3, a Golgi-resident multiple transmembrane protein, generating a lyso form intermediate [79]. In the second step, a saturated chain is transferred back to the sn2-position. PGAP2, a multiple trans-membrane Golgi protein, is required for the reacylation [80]. Whether PGAP2 is the acyltransferase itself remains unclear.

As described in the Introduction, GPI-APs are typical components of membrane microdomains or rafts on the cell surface. In the outer leaflet of the PM, GPI-APs interact with glyco-sphingolipids and cholesterol forming dynamic membrane microdomains of 20–100 nm. Presumably, microdomains are

royalsocietypublishing.org/journal/rsob    Open Biol. **10**: 190290

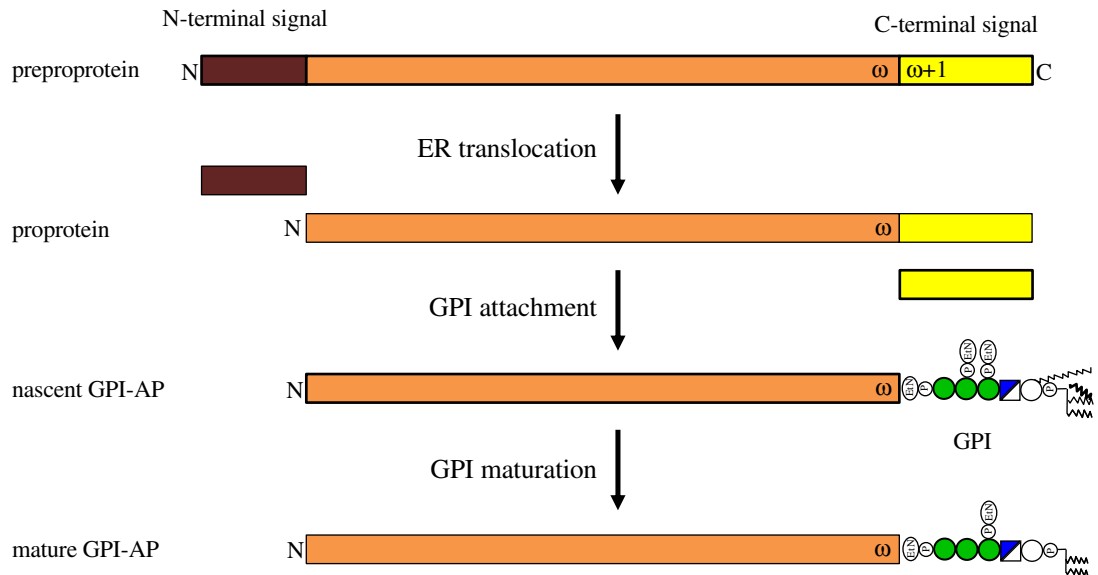

**Figure 4.** Steps in biogenesis of GPI-APs. Preproprotein of GPI-AP has the N-terminal signal peptide for ER localization (brown box) and the C-terminal signal peptide for attachment of GPI (yellow box). The ω site is the amino acid residue, to which GPI is attached. Upon translocation into the ER, the N-terminal signal peptide is cleaved off, generating proprotein. Preassembled GPI is attached to the ω site by replacing the C-terminal signal peptide by GPI transamidase, generating nascent GPI-AP. Nascent GPI-AP undergoes maturation reactions to become mature GPI-AP.

formed through lipid–lipid interactions. The fatty acid remodelling that forms GPI-APs bearing two saturated fatty chains is critical for raft association of GPI-APs [79].

## 3.3. Side chain modification

In many GPI-APs, βGalNAc is transferred to the 4-position of Man1 as a side chain [81]. This modification occurs in the Golgi after fatty acid remodelling (figure 3) [82]. The GalNAc side chain can be elongated by β1–3Gal and Sia (figure 3) [81]. Certain GPI-APs, such as Thy1 [4], always have non-elongated GalNAc as a side chain whereas other GPI-APs, such as the prion and dipeptidase, have mixed GalNAc side chains containing those without elongation and those elongated by Gal or by Gal and Sia [83,84]. The physiological and functional roles of the GalNAc side chains have remained largely unknown. For the prion, a study reported by one group indicated that Sia in the side chain is involved in the conversion of PrP$^c$ to pathogenic PrP$^{sc}$ [85].

Transfer of GalNAc is mediated by PGAP4/TMEM246, a Golgi-resident GPI-specific GalNAc transferase [82]. PGAP4 is widely expressed in various tissues/organs with higher expression in the brain. All Golgi-resident glycosyltransferases (GTs) are type 2 single transmembrane proteins (i.e. they have a short cytoplasmic segment at the N-terminus, followed by a transmembrane domain and a luminal catalytic GT domain [86,87]). By contrast, PGAP4 has three transmembrane domains [82]. Similar to other type 2 GTs, PGAP4 has a short cytoplasmic segment at the N-terminus that is followed by the first transmembrane domain and a GT domain. The GT domain of PGAP4, of the GT-A type, is split into two regions, and between the regions there are tandem transmembrane domains linked by a short stretch of hydrophilic residues. The tandem transmembrane domains inserted into the GT domain locate the GT domains close to the membrane, which may facilitate the interaction of the GT domain with the substrate GPI, which is inserted into the membrane. The

tandem transmembrane domain might also be involved in binding the lipid moiety of the GPI [82].

Recently, the Gal transferase was identified to be B3GALT4 [88], the Golgi-resident Gal transferase previously known to synthesize GM1 from GM2 [89,90]. B3GALT4 also synthesizes GA1 and GD1 from GA2 and GD2, respectively [89]. In these substrate glycosphingolipids, position 3 of GalNAc β1–4-linked to Gal in the lactosylceramide (LacCer) moiety is the acceptor for βGal. Similarly, position 3 of GalNAc β1–4-linked to Man1 is the acceptor in the GPI for βGal. Whereas UDP-Gal and GM2 are sufficient for B3GALT4 to generate GM1, the presence of LacCer is required for Gal transfer to the βGalNAc side chain of the GPI [88]. Because LacCer is a common partial structure of the acceptor substrates of B3GALT4 (GM2, GA2 and GD2), LacCer might directly bind to B3GALT4 and might modulate substrate specificity towards GPI. Sia transferase mediating Sia transfer to the Gal-GalNAc-side chain of the GPI has not been identified.

## 4. Free, non-protein-linked GPI

Protozoan parasites such as *Toxoplasma gondii*, *Plasmodium falciparum*, *Trypanosoma cruzi* and *Leishmania* species have non-protein-linked GPIs as free glycolipids on the cell surface (see the reviews for more details [81,91]). In mammalian cells, there have been few reports about the expression of the unlinked GPI on the cultured cell surface [92–95]. Recently, this issue was revisited [96] using a monoclonal antibody T5_4E10 that was originally generated against *T. gondii* free GPI [97]. T5_4E10 mAb recognizes the non-protein-linked GPI bearing the Man1-linked GalNAc side chain without Gal elongation [82]. Because the T5_4E10 antibody does not bind to the protein linked GPI, it is useful to detect the free GPI bearing non-elongated GalNAc side chain (free GPI-GalNAc from now on) in mammalian cells by flow cytometry or western blotting [96]. Relatively higher levels of free GPI-GalNAc were expressed in the pons, medulla oblongata,

royalsocietypublishing.org/journal/rsob    Open Biol. 10: 190290

**Table 2.** Mammalian proteins involved in GPI -AP biogenesis.

| step | protein | function | step | protein | function |
|---|---|---|---|---|---|
| 1 | PIGA | GPI-GnT, catalytic | 13 | PIGK | GPI transamidase, catalytic 1 |
| 1 | PIGC | GPI-GnT | 13 | GPAA1 | GPI transamidase, catalytic 2 |
| 1 | PIGH | GPI-GnT | 13 | PIGS | GPI transamidase |
| 1 | PIGQ | GPI-GnT | 13 | PIGT | GPI transamidase |
| 1 | PIGP | GPI-GnT | 13 | PIGU | GPI transamidase |
| 1 | PIGY | GPI-GnT | 14 | PGAP1 | inositol-deacylase |
| 1 | DPM2 | GPI-GnT, regulatory | 15 | PGAP5/MPPE1 | EtNP phosphodiesterase |
| 2 | PIGL | GlcNAc-PI-deacetylase | 16 | p24α2 | cargo receptor |
| 3 | ? | flipping | 16 | p24β1 | cargo receptor |
| 4 | PIGW | GlcN-PI-acyltransferase | 16 | p24γ2 | cargo receptor |
| 5 | ? | lipid remodelling | 16 | p24δ1 | cargo receptor |
| 6 | PIGM | GPI-MTI, catalytic | 17 | PGAP3 | fatty acid remodelling, PLA2 |
| 6 | PIGX | GPI-MTI, regulatory | 18 | PGAP2 | fatty acid remodelling, reacylation |
| 7 | PIGV | GPI-MTII | 19 | PGAP4/TMEM246 | GPI-GalNAcT |
| 8 | PIGN | GPI-ETI | 20 | B3GALT4 | GPI-GalT |
| 9 | PIGB | GPI-MTIII | 21 | ? | GPI-SiaT |
| 10 | PIGO | GPI-ETIII, catalytic | | | |
| 10/11 | PIGF | GPI-ETII/III, regulatory | | | |
| 11 | PIGG | GPIETII, catalytic | | | |
| 12 | PIGZ | GPI-MTIV | | | |

spinal cord, testis, epididymis and kidney of adult mice and Neuro2a and CHO cells [96]. In cells defective in GPI transamidase, high levels of free GPI-GalNAc are expressed on the cell surface. Studies using mutant CHO cells, defective in GPI transamidase and one of the genes involved in GPI maturation reactions, demonstrated that free GPIs follow the same structural remodelling pathway as do protein linked GPIs [96]. Therefore, non-protein-linked GPIs exist as glycolipids of some mammalian cell membranes. The physiological roles of the free GPIs are yet to be clarified. By contrast, the pathological effects of abnormally accumulated free GPIs in cells defective in GPI transamidase have been demonstrated in patients with PIGT mutations (see below) [98].

## 5. Comparison of mammalian and yeast GPI biosynthesis

In yeast *S. cerevisiae*, the GPI undergoes two types of lipid remodelling reaction that occur in the ER after attachment to proteins. Fatty acid remodelling of the GPI occurs to the sn2-linked acyl chain of diacyl PI, in that the C16/C18 chain is exchanged with the C26 chain by PER1- and GUP1-dependent reactions [99,100]. PER1 is orthologous to PGAP3 [79], whereas GUP1 is not orthologous to PGAP2 but is a member of the MBOAT family of acyl transferases [100]. The other lipid remodelling of the GPI is its change from a diacylglycerol form to a ceramide form. CWH43 is required for this lipid remodelling [101,102]. The ceramide form of the GPI is not known in mammalian cells; however, parts of CWH43 are homologous to two mammalian proteins. The N-terminal part of CWH43 corresponds to

mammalian PGAP2 [101,102], whereas the C-terminal part of CWH43 corresponds to the PGAP2-interacting protein. Although the latter is termed the PGAP2-interacting protein, there is no evidence for interaction with PGAP2. PGAP2 is localized in the Golgi whereas the PGAP2-interacting protein is localized in the ER [80]. Whether the PGAP2-interacting protein has any functional relation to mammalian GPI biogenesis, for example, involvement in lipid remodelling at the stage of GlcN-(acyl)PI, is yet to be determined.

Yeast defective in ARV1 (for ARE2 required for viability 1) accumulates GlcN-(acyl)PI, suggesting that the first mannosylation of GPI is defective [103]. ARV1 is an ER membrane protein involved in the homeostasis of sterols [104] and sphingolipids [105]. Proper conditions for Dol-P-Man-dependent mannose transfer to GlcN-(acyl)PI might not be generated when ARV1 does not function. Mammalian ARV1 cDNA complemented ARV1-defective yeast, which shows the functional role of mammalian ARV1 [105,106]. Recently, patients with biallelic loss-of-function mutations in ARV1 have been reported [107,108]. The affected children had similar symptoms to those with inherited GPI deficiencies, such as developmental delay and seizures, which is consistent with mammalian ARV1 having a putative role in GPI biosynthesis.

Gpi18p, the yeast PIGV homologue, requires Pga1p for GPI-MTII activity [109]. Gpi18p and Pga1p form a complex. The PGA1 homologue is not found in the mammalian genome. The reason for Gpi18p, but not PIGV, requiring a partner protein is not known.

Yeast has two homologues of PGAP5: TED1 and CDC1. Like PGAP5, TED1 is involved in the removal of EtNP from Man2 [110] whereas CDC1 is required for the removal

of EtNP from Man1 [111]. In mammalian cells, EtNP linked to Man1 remains as a conserved side chain in mature GPI-APs. By contrast, some yeast GPI-APs do not have EtNP on Man1. There is a hypothesis that the removal of EtNP from Man1 is necessary for the incorporation of GPI-APs into the cell wall (see [112] for further discussion).

# 6. Quality control of GPI-APs

When the precursor proteins of GPI-APs are not processed by GPI transamidase for GPI attachment, they are retrotranslocated for degradation by ubiquitin- and proteasome-dependent ER-associated degradation (ERAD) [113]. It is likely that the GPI attachment signal sequence is recognized as an unfolded region, which leads to ubiquitination.

By contrast, when protein folding fails after attachment of the GPI, the misfolded GPI-APs are mainly transported out of the ER for degradation in the lysosomes [114,115]. The process is termed RESET (for rapid ER stress-induced export) [114]. When the ER stress is induced by thapsigargin, the RESET pathway functions efficiently. The ER exit of the misfolded GPI-APs is facilitated by p24 family proteins, which act as cargo receptors of GPI-APs. The misfolded GPI-APs appear transiently on the cell surface before degradation [114]. During the transport through the secretory pathway, misfolded GPI-APs are escorted by ER-derived chaperones, such as BiP and calnexin, and p24 proteins [116]. Most GPI-APs have at least one N-glycan and their folding is mediated by N-glycan-dependent calnexin cycle [117]. Calnexin also binds to PGAP1 and facilitates PGAP1-mediated removal of the acyl chain from inositol [117]. When the calnexin cycle is inefficient, GPI-APs bearing inositol-linked acyl chain appear on the cell surface.

In yeast, misfolded GPI-APs are targeted to ERAD to some extent, while a sizable fraction of the misfolded GPI-APs exit the ER for degradation in vacuoles [115,118]. For efficient ER–Golgi transport of GPI-APs mediated by p24 cargo receptors, remodelling of both lipid and glycan moieties is required [119]. These GPI remodelling reactions occur even when the protein moiety is misfolded, allowing efficient recruitment into COPII-coated transport vesicles. In this regard, quality control of GPI-APs in the ER is limited and the misfolded GPI-APs pass other compartments in the secretory pathway [115].

# 7. Shedding of GPI-APs mediated by GPI cleaving/processing enzymes (GPIases)

A prominent characteristic of GPI-APs is their shedding from the cell surface. There are several pairs of GPI-APs and GPI cleaving/processing enzymes, GPIases, that are responsible for their shedding. RECK (for reversion-inducing-cysteine-rich protein with Kazal motifs) is a GPI-anchored inhibitor of matrix metalloproteinases, such as MMP-9 and ADAM10 [120]. GDE2 is a member of the glycerophosphodiester phosphodiesterase family, having six transmembrane domains and an extracellular phosphodiester domain [121]. There is evidence that GDE2 has RECK-specific GPI phospholipase C activity [26]. RECK suppresses ADAM10 on certain neuronal progenitor cells [122]. When RECK is released by GDE2, ADAM10 cleaves a Notch ligand, such as the Delta-like 1, on the same cell, which in turn shuts down Notch signalling into the neighbour neuronal progenitor cell, leading to initiation of neuronal differentiation [26,123].

The urokinase receptor (uPAR), a GPI-AP, mediates degradation of the extracellular matrix through protease recruitment and enhances cell adhesion, migration and invasion. Cell surface activity of the uPAR is negatively regulated by its shedding from the cell surface, which is mediated by GDE3, another member of the glycerophosphodiester phosphodiesterase family [27]. GDE3 mediates cleavage and release of the uPAR by its GPI-specific phospholipase C activity. Interestingly, the uPAR is resistant to GDE2 [27], suggesting that substrate specificities of GDE2 and GDE3 are dependent upon the protein portions of RECK and the uPAR, respectively.

PGAP6/TMEM8A is a GPI-specific phospholipase A2 expressed on the surface of some cells, including embryonic cells. PGAP6 has sequence similarity to PGAP3, a Golgi-resident GPI-specific phospholipase A2 involved in fatty acid remodelling of the GPI [28]. PGAP6 and PGAP3 belong to the CREST (for alkaline ceramidase, PAQR receptor, Per1, SID-1 and TMEM8) superfamily of hydrolases [124]. The substrate of PGAP6 is CRIPTO-1 [28], a GPI-anchored co-receptor of TGFβ receptors [125]. CRYPTIC, a GPI-AP closely related to CRIPTO-1, is resistant to PGAP6, suggesting that PGAP6 is a CRIPTO-1-specific phospholipase A2 [28]. When CRIPTO-1 and PGAP6 are expressed on the same cell, CRIPTO-1 is processed by PGAP6's phospholipase A2 activity, loses one fatty acyl chain (lyso form of GPI) and is spontaneously released from the cell surface. When GPI-specific phospholipase D (GPI-PLD) is present in the surrounding medium or is expressed in the cell, the lyso form of CRIPTO-1 is cleaved and more efficiently released [28]. The released CRIPTO-1 possessed its activity as a co-receptor of TGFβ receptors [126]. CRIPTO-1 is involved in various steps in embryogenesis. In an early mouse embryonic stage (6.5 days post coitum), Cripto-1 is essential in the generation of the anterior–posterior axis [127]. A fraction of PGAP6 knockout embryos do not form the anterior–posterior axis, confirming the critical role of PGAP6-dependent shedding of CRIPTO-1 in Nodal signalling regulation *in vivo* [28].

A complex of two GPI-APs, LY6 K and TEX101, is required for sperm migration into the oviduct. Males of LY6 K knockout mice and TEX101 knockout mice are infertile. Their sperm are defective in binding to an egg's zona pellucida *in vitro* as well as in migration into the oviduct *in vivo* [128]. A testis form of angiotensin-converting enzyme (ACE) is essential for the maturation of spermatids. Similar to LY6 K knockout and TEX101 knockout male mice, ACE knockout male mice are infertile and their sperm does not pass through the uterotubal junction *in vivo* and does not bind to the zona pellucida *in vitro*. ACE cleaves GPI by its putative endomannosidase activity, which seems to reside in a different site from the well-characterized carboxy dipeptidase catalytic site for conversion of angiotensin [129]. TEX101 is a specific target for the GPIase activity of ACE [29]. Shedding of TEX101 by ACE is critical for spermatozoa to become fertilization competent [29,128].

Pairs of the GPI-AP substrates and GPIases described above are examples of the protein specific cleavage/processing of the GPI. By contrast, GPI-PLD has been known for a long time as a GPIase that is active against a wide variety of GPI-APs [130,131]. However, GPI-PLD requires a detergent for its GPIase activity towards the cell surface GPI-APs [132].

royalsocietypublishing.org/journal/rsob    Open Biol. 10: 190290

As described above, GPI-PLD cleaves the lyso form of GPI-APs in the absence of a detergent, facilitating their shedding from the cell surface [28]. Further, GPI-PLD is able to cleave GPI-APs within the intracellular secretory pathway [133]. These results suggest that when GPI-APs are associated with the membrane microdomains, they are resistant to GPI-PLD. Studies with GPI-PLD knockout mice suggested that GPI-PLD generates diacylglycerol by cleaving GPI within hepatocytes where GPI-PLD expression is most abundant [134]. Diacylglycerol might activate PKCε, which in turn binds to the insulin receptor and inhibits its tyrosine kinase activity, leading to insulin resistance. GPI-PLD knockout ameliorated glucose intolerance and hepatic steatosis under a high-fat and high-sucrose diet through a reduction of diacylglycerol and a subsequent decrease of PKCε activity [134].

# 8. Transcytotic endocytosis of GPI-anchored receptors

Several GPI-anchored receptors act as transcytotic receptors that transport ligands from one membrane domain to another in endothelial and epithelial cells. GPIHBP1 (GPI-anchored heparin-binding protein 1) binds lipoprotein lipase (LPL) on the basolateral surface of capillary endothelial cells and transport it to the apical (vascular) surface [23]. On the apical surface of capillary endothelial cells, LPL acts on lipoproteins to liberate fatty acids to fuel tissues (see the review for further discussion [135]). Folate receptor 1, a GPI-anchored type of folate receptors, binds 5-methyltetrahydrofolate (5MHF) on the vascular (basolateral) side of the epithelial cells of the choroid plexus in the brain and is endocytosed. The folate receptor 1 and 5MHF complexes are released from the apical surface into the cerebrospinal fluid as exosomes, which then pass the ependyma and enter the brain parenchyma. This is a critical route of 5MHF incorporation from the blood to the brain parenchyma [24]. Glycoprotein 2 (GP2) is selectively expressed on the surface of M cells, cells in the mucosal lymphoid tissue Peyer's patch. GP2 is involved in sampling foreign antigens into the mucosal immune system (see the review for further discussion [136]). Botulinum toxin produced by the food-borne botulism-causing bacteria *Clostridium botulinum* in the intestine binds to GP2 on the apical surface of M cells and traverses the intestinal epithelial layer via transcytosis [25].

# 9. Lessons from GPI deficiencies

## 9.1. Molecular genetics of GPI deficiencies

Defective biosynthesis of the GPI is caused by several different genetic mechanisms. The first GPI deficiency that was clarified at the molecular genetics level is paroxysmal nocturnal haemoglobinuria (PNH) (table 3) [137]. GPI deficiency in PNH is caused by somatic mutations in PIGA that occur in the haematopoietic stem cells (HSCs) [138]. PIGA is the only X-linked gene among all genes involved in GPI biosynthesis. Because of its X-linkage, one loss-of-function somatic mutation in PIGA causes a GPI defective phenotype [138]. This is true for both male and female cells because one X chromosome in female cells was inactivated during embryogenesis and hence a mutation in PIGA in the active X chromosome is sufficient to cause the phenotype. Because PIGA mutation exists only in the affected blood cells but not in germ cells, PNH is not an inherited disease. PIGA somatic mutations found in patients with PNH cause complete loss or a severe reduction of GPI-APs on the cell surface. It should be pointed out that the somatic PIGA mutation alone does not cause PNH because the generation of one GPI-AP-defective HSC among many hundreds of HSCs should not affect the blood system. Indeed, similar PIGA somatic mutations are found in blood granulocytes from healthy individuals [139]. When the mutant haematopoietic stem cell exhibits clonal expansion and generates large numbers of GPI-AP-defective blood cells, clinical PNH is manifested (see reviews for further discussion about clonal expansion [137]).

Another form of GPI deficiency is inherited GPI deficiency (IGD) (table 3). The first inherited form of GPI deficiency was reported in 2006 [140]. A homozygous hypomorphic mutation in PIGM located in chromosome 1q was found in three children from two consanguineous families. The affected children suffered from seizures and thrombosis in the hepatic or portal vein. The mutation was within the promoter region of the PIGM gene and disrupted an Sp1-binding site important for PIGM transcription in some cell types, such as blood granulocytes, B-lymphocytes and fibroblasts. In those cells, the cell surface levels of some GPI-APs are reduced.

In 2010 when the application of whole-exome sequencing to rare inherited diseases became feasible, biallelic (homozygous or compound heterozygous) mutations in PIGV located in chromosome 1p were found in several affected children with Mabry syndrome, also known as hyperphosphatasia with mental retardation syndrome (HPMRS) [141]. The affected children inherited loss-of-function mutations from their parents who were heterozygotes of wild-type and mutant alleles. GPI-APs in blood cells and fibroblasts from affected individuals were partially lost by these biallelic PIGV mutations.

To date, pathogenic germ line mutations causing IGD have been found in 21 out of 27 genes in GPI biosynthesis [140–157] and the GPI-AP maturation pathway [158–161] (table 3). Inheritance of IGDs caused by mutations in autosomal genes, such as PIGB-IGD [151] and PIGC-IGD [147], is autosomal recessive. Inheritance of IGD caused by germ line mutations of PIGA (PIGA-IGD) is X-linked recessive, in which all affected individuals are boys [145]. Their mothers who were heterozygous carriers of a PIGA mutation, were phenotypically mosaic at the cell level (blood cells were mixtures of normal and GPI-AP-deficient cells) but did not have clear symptoms of IGD [162].

More recently, cases of PNH caused by mutations of PIGT were found [98,163,164]. These cases did not have the somatic PIGA mutation found in the affected blood cells from patients with PNH, but instead had biallelic loss-of-function mutations of the PIGT gene. One PIGT allele had germ line mutations whereas the other PIGT was lost by deletion of the 8–24 Mb region of chromosome 20q, which somatically occurred in a haematopoietic stem cell [98]. Therefore, GPI-APs were lost by a combination of germ line mutation and somatic mutation. The somatic mutation is always caused by a Mb scale deletion because simultaneous deletion of the myeloid common deleted region [98], implicated in clonal expansion of the HSCs [165,166], is required for clinical manifestation of PIGT-PNH.

**Table 3.** Diseases caused by loss-of-function mutations in PIG and PGAP genes.

| step | gene | disease | symptoms | patients[a] | Chr[b] | mutation |
|------|------|---------|----------|-------------|--------|----------|
| 1 | PIGA | PNH[c] | haemolysis, thrombosis | many | Xp | S[d] in HSC[e] |
| | | IGD[f]/MCAHS[g] | Sz[h], DD/ID[i], Hpt[j] | 26 | | G[k] |
| 1 | PIGC | IGD | Sz, DD/ID, Hpt | 2 | 1q | G |
| 1 | PIGH | IGD | Sz, DD/ID, Hpt | 2 | 14q | G |
| 1 | PIGP | IGD | Sz, DD/ID, Hpt | 2 | 21q | G |
| 1 | PIGQ | IGD | Sz, DD/ID, Hpt | 3 | 16p | G |
| 1 | PIGY | IGD | Sz, DD/ID, Hpt | 4 | 4q | G |
| 2 | PIGL | IGD, CHIME syndrome | Sz, DD/ID, Hpt | 15 | 17p | G |
| 4 | PIGW | IGD/HPMRS[l] | Sz, DD/ID, Hpt | 3 | 17q | G |
| 6 | PIGM | IGD | Sz, thrombosis | 7 | 1q | G, promotor |
| 7 | PIGV | IGD/HPMRS | Sz, DD/ID, Hpt | 18 | 1p | G |
| 8 | PIGN | IGD/MCAHS, Fryns syndrome | Sz, DD/ID, Hpt | 22 | 18q | G |
| 9 | PIGB | IGD/HPMRS | Sz, DD/ID, Hpt | 12 | 15q | G |
| 10 | PIGO | IGD/HPMRS | Sz, DD/ID, Hpt | 13 | 9p | G |
| 11 | PIGG | IGD | Sz, DD/ID, Hpt | 7 | 4p | G |
| 13 | GPAA1 | IGD | Sz, DD/ID, Hpt | 10 | 8q | G |
| 13 | PIGS | IGD | Sz, DD/ID, Hpt | 7 | 17p | G |
| 13 | PIGT | PIGT-PNH | haemolysis, thrombosis, infla[m] | 4 | 20q | G + S in HSC |
| | | IGD/MCAHS | Sz, DD/ID, Hpt | 28 | | G |
| 13 | PIGU | IGD | Sz, DD/ID, Hpt | 5 | 20q | G |
| 14 | PGAP1 | IGD | Sz, DD/ID, Hpt | 8 | 2q | G |
| 17 | PGAP3 | IGD/HPMRS | Sz, DD/ID, Hpt | 45 | 17q | G |
| 18 | PGAP2 | IGD/HPMRS | Sz, DD/ID, Hpt | 23 | 11p | G |

[a]Numbers of published patients as of December 2019.
[b]Chr, chromosome location.
[c]PNH, paroxysmal nocturnal haemoglobinuria.
[d]S, somatic mutations.
[e]HSC, haematopoietic stem cell.
[f]IGD, inherited GPI deficiency.
[g]MCAHS, multiple congenital anomalies-hypotonia-seizures syndrome.
[h]Sz, seizures.
[i]DD/ID, developmental delay/intellectual disability.
[j]Hpt, hypotonia.
[k]G, germline mutations.
[l]HPMRS, hyperphosphatasia mental retardation syndrome/Mabry syndrome.
[m]infla, inflammation.

## 9.2. Genotype–phenotype relationship

Twenty-one genes whose mutations caused IGD cover almost all steps in the biosynthesis of the core GPI, its transfer to proteins and maturation of the GPI, for which genes are known except a step mediated by PGAP5 [140–146,149,151,153,158,161]. It seems, therefore, that every step in GPI-AP biogenesis is critical for human health. There has been no report of pathogenic mutations in genes involved in side chain modifications (i.e. PIGZ and PGAP4); therefore, it is unknown what roles the side chains of the GPI play in human health.

Complete loss of GPI biosynthesis in the whole body of the mouse caused early embryonic lethality, as demonstrated by knockout of the Piga gene [167]. Complete GPI deficiency would also cause a similar phenotype in a human being. A partial but severe reduction in GPI biosynthesis would also cause embryonic lethality. A homozygous PIGC mutation associated with a family with recurrent foetal loss seems highly likely to be such a case [168]. If the reduction in GPI biosynthesis is less severe, the outcome would be IGD. In fact, hypomorphic mutations of PIGC have been associated with epilepsy and intellectual disability [147]. As summarized in recent articles, the most prominent clinical symptoms of IGD are neurological ones, including seizures, developmental delay/intellectual disability, cerebral atrophy and hypotonia [169,170]. Many GPI-APs, such as contactins, tissue non-specific alkaline phosphatase (TNAP) and Thy1, are expressed on neurons, oligodendrocytes and other glial cells. Partial reduction of these GPI-APs impairs neurological development and functions, highlighting the importance of GPI-APs in the neuronal system. One example is a causal

royalsocietypublishing.org/journal/rsob    Open Biol. 10: 190290

role of defective expression of TNAP in seizures [171,172]. TNAP is involved in cellular uptake of pyridoxal phosphate, active vitamin B6, where TNAP dephosphorylates pyridoxal phosphate to generate membrane permeable pyridoxal. Once in the cytoplasm, pyridoxal is reverted to pyridoxal phosphate by pyridoxal kinase and is used by many B6-dependent enzymes including GABA (γ-aminobutyric acid) synthetic enzyme. Therefore, the reduction of cell surface TNAP might result in reduced GABA synthesis, leading to the easy occurrence of seizures.

Concerning genes required for the biosynthesis of the core GPI, many mutations found in individuals with IGD are null or nearly null [173]. Heterozygotes of such null alleles in the same families are healthy, suggesting that 50% of normal levels of GPI biosynthesis may be sufficient for not causing clinical symptoms. When the null allele is combined with a hypomorphic allele, GPI levels further decrease, resulting in symptoms of IGD.

Concerning the PGAP1 and PGAP3 genes involved in the maturation of the GPI, homozygous null alleles do not cause embryonic lethality, but do cause IGD [158,161]. The cell surface levels of GPI-APs do not decrease, or only slightly decrease, in PGAP1 or PGAP3 knockout cells, whereas structures of GPI moiety in GPI-APs are different from those in wild-type cells. In PGAP1 knockout cells, the inositol-linked acyl chain remains in GPI-APs [71]. In PGAP3 knockout cells, the fatty acid remodelling does not occur (i.e. the sn2-linked fatty acid remains as unsaturated fatty acid [79]). These structural abnormalities are probably causally related to the functional impairment of certain GPI-APs, resulting in clinical symptoms. Recently, it was demonstrated that fatty acid remodelling of GPI-APs is required for the generation of nanoclusters of GPI-APs, which is critical for activation of β1-integrins on lymphocytes and their subsequent spreading and migration [174]. The role of GPI fatty acid remodelling in cellular physiology might be causally related to clinical symptoms of IGD caused by PGAP3 mutations.

Concerning genes of GPI transamidase, such as PIGT, mutations that cause complete or nearly complete loss of GPI transamidase activity of a cell lead not only to complete or nearly complete lack of GPI-APs but also to the accumulation of unlinked free GPI in the cell. By contrast, mutations that cause partial loss of GPI transamidase activity lead to partial reduction of cell surface GPI-APs. The same germ line PIGT mutation has been found in IGD [175,176] and PNH [164]. PIGT bearing c.250G>T, p.E84X is a genetic variant found in the East Asian population at an allele frequency of 0.00023. PIGT E84X has very weak activity due to read through of the nonsense codon. Two Japanese children with IGD, who had no familial relationship, inherited this variant PIGT and different PIGT variants from their parents. Both of the other variant PIGTs had partially reduced activities [175,176]. Their cells had subnormal levels of GPI-APs [175,176]. A Japanese adult patient with PIGT-PNH had the same E84X variant in the germ line [164]. In his clonal PNH blood cells, the wild-type allele of PIGT was lost somatically as described above. This combination of a nearly null germ line mutation and somatic loss of the normal allele caused a nearly complete lack of GPI-APs, resulting in PNH [164]. The two children with IGD did not have PNH symptoms, such as intravascular haemolysis, because their blood cells had nearly normal levels of GPI-anchored complement regulators, CD59 and CD55 [175,176].

Patients with PIGT-PNH had typical complement-mediated intravascular haemolysis and, in addition, recurrent autoinflammatory symptoms, such as urticaria, arthralgia and noninfectious meningitis, associated with elevated IL18 and serum amyloid A [98]. Unlinked free GPIs were expressed on affected erythrocytes, monocytes, granulocytes and B-lymphocytes. Both intravascular haemolysis and autoinflammatory symptoms were controlled by anti-complement C5 antibody drug eculizumab, suggesting the involvement of terminal complement activation for both. Studies comparing PIGT- and PIGA-knockout monocyte/macrophage cell lines showed that accumulated free GPI on PIGT-knockout cells caused enhanced complement activation and subsequent IL1β secretion. Therefore, although free GPI is normal component of certain cells, abnormally accumulated free GPI is pathogenic. It is conceivable that when free GPI levels are abnormally high, interactions with some unknown protein(s) involved in complement activation, such as lectin pathway components, may be enhanced due to multivalent binding.

## 10. Concluding remarks

Biogenesis of mammalian GPI-APs have been studied well, but some points remain to be clarified: mechanisms of ER translocation of preproteins of mammalian GPI-APs; genes involved in flip of GlcN-PI into the luminal side of the ER; mechanisms and genes involved in lipid remodelling of GlcN-(acyl)PI; identification of the sialyl transferase for elongation of GalNAc-Gal side chain; and functions of individual components of GPI-GnT and GPI transamidase. Physiological roles of Man4 and GalNAc side chains need to be studied. Physiological roles of unlinked free GPI found in some tissues by T5_4E10 monoclonal antibody need to be understood. Because T5_4E10 antibody binds only to free GPI bearing GalNAc as a side chain, probes for other forms of potentially expressed free GPIs, such as those lacking a side chain and those bearing GalNAc side chain with elongation by Gal and Sia, are required to elucidate an entire picture of free GPIs. There might be more examples of biologically important GPI-AP shedding mediated by GPIases that are specific to the target GPI-APs.

Data accessibility. This article has no additional data.

Competing interests. I declare I have no competing interests.

Funding. This work was funded by Ministry of Education, Culture, Sports, Science and Technology (grant no. KAKENHI 17H06422).

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
