## [Reviewer comments · Open Biology]

Review History

RSOB-19-0290.R0 (Original submission)

Review form: Reviewer 1

Recommendation

Accept with minor revision (please list in comments)

Do you have any ethical concerns with this paper?

No

Comments to the Author

This is an excellent review that nicely summarizes the current knowledge of the biosynthesis and maturation of GPI-anchored proteins (GPI-APs) in mammalian cells, highlighting the importance of the different steps of these processes in the trafficking and functions of this class of proteins. Interestingly, the author also examines the unique characteristics of GPI-APs (such as association with lipid domains, cell surface shedding, etc.) through specific examples, emphasizing the physiological relevance. Finally, the article also nicely reviews the recent findings on the inherited GPI deficiency caused by mutations in genes involved in the GPI-AP biosynthesis and maturation, critically pointing out the genotype-phenotype relationship. Overall, the review is very informative; it is well written, the figures and tables are clear and appropriate.

I have just few suggestions:

1) Author discussed that the association with lipid domains is a prominent feature of GPI-APs, describing some properties of GPI-AP enriched domains. However, while the author highlighted the role of lipid-lipid and lipid-glycan interactions, he missed the critical role of protein-protein interactions in regulating formation and maintenance of GPI-AP nanodomains (Suzuky et al, Nat Chem Biol 2012; Paladino et al, Nat Chem Biol 2014; Lebreton et al, Crit Rev in Biochem Mol Biol 2018). It would be important to debate this concept.

2) For reviews for mechanisms of GPI-APs sorting in polarized epithelial cells (pag. 4), I suggest to cite more recent ones that include the current advances in the field:

Zurzolo and Simons. Glycosylphosphatidylinositol-anchored proteins. Membrane organization and transport. BBA 2016;

Lebreton, Paladino, Zurzolo. Clustering in the Golgi apparatus governs sorting and function of GPI-APs in polarized epithelial cells, Febs Letters 2019

3) At pag. 14: the author wrote "the presence of LacCer is required for Gal transfer to the β GalNAc side chain of the GPI", this sentence is too reductive and it is not clear for the reader how the LacCer is critical for the remodeling of GPI-anchor. Moreover, these data imply an intriguing relationship between GPI-AP biosynthesis/remodeling and glycosphingolipid biosynthesis. It would be important to debate this new concept.

Minor comment:

For clarity, spell "CSF" (pag 22, line 466)

Review form: Reviewer 2

Recommendation

Accept with minor revision (please list in comments)

Do you have any ethical concerns with this paper?

No

Comments to the Author

This is a comprehensive and authoritative review, that any student of modern cell biology with an interest in physiology must read.

This is a comprehensive and authoritative review, that any student of modern cell biology with an interest in cell and organismal physiology must read. The biosynthesis parts have been written with deep knowledge and simplicity that make this complex process accessible to the lay reader. The authors have highlighted many peculiar aspects of clinical phenotypes that occur due to defects in GPI-anchor biosynthesis. I have no reservation whatsoever about the outstanding quality of this review and it must be published soon.

Some suggestions for improvement:

1) A criticism if any, is the lack of mechanistic insight in the generation of the clinical phenotypes that occur due to the mutations in most places. It would be very valuable to have the perspective of one the few people who study this process (biosynthesis of GPI), and is also a leading clinician with an abiding interest in disease.

i) For example: on pg. 16 lines 337- 342 that authors state- ' Mammalian ARV1 cDNA

complemented ARV1-defective yeast, which shows the functional role of mammalian ARV1 (101, 102). Recently, patients with biallelic loss-of-function mutations in ARV1 have been reported (103, 104). The affected children had similar symptoms to those with inherited GPI deficiencies, such as developmental delay and seizures, which is consistent with mammalian ARV1 having a putative role in GPI biosynthesis.' However neither here nor anywhere else do we understand how GPI-biosynthesis relates to developmental delays and seizures.

ii) Or- on page 23: The authors blandly state 'GPI-PLD knockout ameliorated glucose intolerance and hepatic steatosis under a high-fat and high-sucrose diet through a reduction of diacylglycerol and a subsequent decrease of PKC ϵ activity (130).'

iii) or on page 26: A mystery that has persisted is why PIGA mutations only cause PNH defects when they occur in stem cells, and that the somatic PIGA mutation alone does not cause PNH. 'Indeed, similar PIGA somatic mutations are found in blood granulocytes from healthy individuals (135). When the mutant hematopoietic stem cell exhibits clonal expansion and generates large numbers of GPI-AP-defective blood cells, clinical PNH is manifested'??

iv) The authors state: line 596 Therefore, although free GPI is normal component of certain cells, abnormally accumulated free GPI is pathogenic.' This has also been repeated in an earlier section lines 310-313. Again, this is a very intriguing result and could do with some mechanistic insight from the author.

In all these cases perhaps it's too premature for the author to speculate, but some insights about mechanism from the deep experience of the author would be valuable to pen down.

2) In Table 2: it would be useful to have a table of diseases or phenotypes associated with the mutations- or have a separate table that could delineate this.

3) In Figures 2) and 3) it would be useful to map the names of the genes outlined in Table 2 to the steps indicated as 1,2,3 and so on to PIGA, M etc, so that there is a ready visual reckoning of the different genes to the steps that are involved in GPI biosynthesis.

4) Recently, a role for PGAP2 and 3 PIGX and PIGM, respectively, in integrin signalling has been identified. This study could provide a functional insight into how the cell may use the remodelled fatty acids on GPI anchored proteins to make functional membrane microdomains (see Kalappurakkal et al. Cell. 2019 Jun 13;177(7):1738-1756). This is an interesting insight into how PIGX and PIGM deletions may affect organismal physiology, and may be discussed here.

Minor:

Line 344: ..Pga1p should read PGA1p.

Line 544 'embryonic lethality, as demonstrated by knockout of the Piga gene (163).' It should read...PIGA.

Decision letter (RSOB-19-0290.R0)

10-Jan-2020

Dear Professor Kinoshita,

We are pleased to inform you that your manuscript RSOB-19-0290 entitled "Biosynthesis and biology of mammalian GPI-anchored proteins" has been accepted by the Editor for publication in Open Biology. The reviewer(s) have recommended publication, but also suggest some minor

revisions to your manuscript. Therefore, we invite you to respond to the reviewer(s)' comments and revise your manuscript.

Please submit the revised version of your manuscript within 7 days. If you do not think you will be able to meet this date please let us know immediately and we can extend this deadline for you.

- 1) A text file of the manuscript (doc, txt, rtf or tex), including the references, tables (including captions) and figure captions. Please remove any tracked changes from the text before submission. PDF files are not an accepted format for the "Main Document".
- 2) A separate electronic file of each figure (tiff, EPS or print-quality PDF preferred). The format should be produced directly from original creation package, or original software format. Please note that PowerPoint files are not accepted.
- 3) Electronic supplementary material: this should be contained in a separate file from the main text and meet our ESM criteria (see <https://royalsociety.org/journals/authors/author-guidelines/>). All supplementary materials accompanying an accepted article will be treated as in their final form. They will be published alongside the paper on the journal website and posted on the online figshare repository. Files on figshare will be made available approximately one week before the accompanying article so that the supplementary material can be attributed a unique DOI.

Online supplementary material will also carry the title and description provided during submission, so please ensure these are accurate and informative. Note that the Royal Society will not edit or typeset supplementary material and it will be hosted as provided. Please ensure that the supplementary material includes the paper details (authors, title, journal name, article DOI). Your article DOI will be 10.1098/rsob.2016[last 4 digits of e.g. 10.1098/rsob.20160049].

- 4) A media summary: a short non-technical summary (up to 100 words) of the key findings/importance of your manuscript. Please try to write in simple English, avoid jargon, explain the importance of the topic, outline the main implications and describe why this topic is newsworthy.

Images

Data-Sharing

It is a condition of publication that data supporting your paper are made available. Data should be made available either in the electronic supplementary material or through an appropriate repository. Details of how to access data should be included in your paper. Please see <https://royalsociety.org/journals/authors/author-guidelines/> for more details.

Data accessibility section

Sincerely,

The Open Biology Team

<mailto:openbiology@royalsociety.org>

Reviewer(s)' Comments to Author:

Referee: 1

Comments to the Author(s)

This is an excellent review that nicely summarizes the current knowledge of the biosynthesis and maturation of GPI-anchored proteins (GPI-APs) in mammalian cells, highlighting the importance of the different steps of these processes in the trafficking and functions of this class of proteins. Interestingly, the author also examines the unique characteristics of GPI-APs (such as association with lipid domains, cell surface shedding, etc.) through specific examples, emphasizing the physiological relevance. Finally, the article also nicely reviews the recent findings on the inherited GPI deficiency caused by mutations in genes involved in the GPI-AP biosynthesis and maturation, critically pointing out the genotype-phenotype relationship. Overall, the review is very informative; it is well written, the figures and tables are clear and appropriate.

I have just few suggestions:

1) Author discussed that the association with lipid domains is a prominent feature of GPI-APs, describing some properties of GPI-AP enriched domains. However, while the author highlighted the role of lipid-lipid and lipid-glycan interactions, he missed the critical role of protein-protein interactions in regulating formation and maintenance of GPI-AP nanodomains (Suzuky et al, *Nat Chem Biol* 2012; Paladino et al, *Nat Chem Biol* 2014; Lebreton et al, *Crit Rev in Biochem Mol Biol* 2018). It would be important to debate this concept.

2) For reviews for mechanisms of GPI-APs sorting in polarized epithelial cells (pag. 4), I suggest to cite more recent ones that include the current advances in the field:

Zurzolo and Simons. Glycosylphosphatidylinositol-anchored proteins. Membrane organization and transport. *BBA* 2016;

Lebreton, Paladino, Zurzolo. Clustering in the Golgi apparatus governs sorting and function of GPI-APs in polarized epithelial cells, *Febs Letters* 2019

3) At pag. 14: the author wrote “the presence of LacCer is required for Gal transfer to the β GalNAc side chain of the GPI”, this sentence is too reductive and it is not clear for the reader how the LacCer is critical for the remodeling of GPI-anchor. Moreover, these data imply an intriguing relationship between GPI-AP biosynthesis/remodeling and glycosphingolipid biosynthesis. It would be important to debate this new concept.

Minor comment:

For clarity, spell “CSF” (pag 22, line 466)

Referee: 2

Comments to the Author(s)

This is a comprehensive and authoritative review, that any student of modern cell biology with an interest in physiology must read.

This is a comprehensive and authoritative review, that any student of modern cell biology with an interest in cell and organismal physiology must read. The biosynthesis parts have been written with deep knowledge and simplicity that make this complex process accessible to the lay reader. The authors have highlighted many peculiar aspects of clinical phenotypes that occur due to defects in GPI-anchor biosynthesis. I have no reservation whatsoever about the outstanding quality of this review and it must be published soon.

Some suggestions for improvement:

1) A criticism if any, is the lack of mechanistic insight in the generation of the clinical phenotypes that occur due to the mutations in most places. It would be very valuable to have the perspective of one the few people who study this process (biosynthesis of GPI), and is also a leading clinician with an abiding interest in disease.

i) For example: on pg. 16 lines 337- 342 that authors state- ‘ Mammalian ARV1 cDNA complemented ARV1-defective yeast, which shows the functional role of mammalian ARV1 (101, 102). Recently, patients with biallelic loss-of-function mutations in ARV1 have been reported (103, 104). The affected children had similar symptoms to those with inherited GPI deficiencies, such as developmental delay and seizures, which is consistent with mammalian ARV1 having a putative role in GPI biosynthesis.’ However neither here nor anywhere else do we understand how GPI-biosynthesis relates to developmental delays and seizures.

ii) Or- on page 23: The authors blandly state ‘GPI-PLD knockout ameliorated glucose intolerance and hepatic steatosis under a high-fat and high-sucrose diet through a reduction of diacylglycerol and a subsequent decrease of PKC ϵ activity (130).’

iii) or on page 26: A mystery that has persisted is why PIGA mutations only cause PNH defects when they occur in stem cells, and that the somatic PIGA mutation alone does not cause PNH. ‘Indeed, similar PIGA somatic mutations are found in blood granulocytes from healthy individuals (135). When the mutant hematopoietic stem cell exhibits clonal expansion and generates large numbers of GPI-AP-defective blood cells, clinical PNH is manifested’??

iv) The authors state: line 596 Therefore, although free GPI is normal component of certain cells, abnormally accumulated free GPI is pathogenic.’ This has also been repeated in an earlier section lines 310-313. Again, this is a very intriguing result and could do with some mechanistic insight from the author.

In all these cases perhaps it’s too premature for the author to speculate, but some insights about mechanism from the deep experience of the author would be valuable to pen down.

2) In Table 2: it would be useful to have a table of diseases or phenotypes associated with the mutations- or have a separate table that could delineate this.

3) In Figures 2) and 3) it would be useful to map the names of the genes outlined in Table 2 to the steps indicated as 1,2,3 and so on to PIGA, M etc, so that there is a ready visual reckoning of the different genes to the steps that are involved in GPI biosynthesis.

4) Recently, a role for PGAP2 and 3 PIGX and PIGM, respectively, in integrin signalling has been identified. This study could provide a functional insight into how the cell may use the remodelled fatty acids on GPI anchored proteins to make functional membrane microdomains (see Kalappurakkal et al. Cell. 2019 Jun 13;177(7):1738-1756). This is an interesting insight into how PIGX and PIGM deletions may affect organismal physiology, and may be discussed here.

Minor:

Line 344: ..Pga1p should read PGA1p.

Line 544 'embryonic lethality, as demonstrated by knockout of the Piga gene (163).' It should read...PIGA.

Author's Response to Decision Letter for (RSOB-19-0290.R0)

See Appendix A.

Decision letter (RSOB-19-0290.R1)

12-Feb-2020

Dear Professor Kinoshita

We are pleased to inform you that your manuscript entitled "Biosynthesis and biology of mammalian GPI-anchored proteins" has been accepted by the Editor for publication in Open Biology.

Sincerely,

The Open Biology Team

mailto:openbiology@royalsociety.org

Appendix A

February 5, 2020

Dear The Open Biology Team,

RE: RSOB-19-0290

I am submitting a revised version of the above referenced manuscript entitled “Biosynthesis and biology of mammalian GPI-anchored proteins”. I revised the manuscript according to the referees’ suggestions and comments as detailed below.

Thank you very much again for your help.

Taroh Kinoshita

Referee: 1

Comments to the Author(s)

This is an excellent review that nicely summarizes the current knowledge of the biosynthesis and maturation of GPI-anchored proteins (GPI-APs) in mammalian cells, highlighting the importance of the different steps of these processes in the trafficking and functions of this class of proteins. Interestingly, the author also examines the unique characteristics of GPI-APs (such as association with lipid domains, cell surface shedding, etc.) through specific examples, emphasizing the physiological relevance.

Finally, the article also nicely reviews the recent findings on the inherited GPI deficiency caused by mutations in genes involved in the GPI-AP biosynthesis and maturation, critically pointing out the genotype-phenotype relationship.

Overall, the review is very informative; it is well written, the figures and tables are clear and appropriate.

I have just few suggestions:

1) Author discussed that the association with lipid domains is a prominent feature of GPI-APs, describing some properties of GPI-AP enriched domains. However, while the author highlighted the role of lipid-lipid and lipid-glycan interactions, he missed the critical role of protein-protein interactions in regulating formation and maintenance of GPI-AP nanodomains (Suzuky et al, Nat Chem Biol 2012; Paladino et

al, Nat Chem Biol 2014; Lebreton et al, Crit Rev in Biochem Mol Biol 2018). It would be important to debate this concept.

Thank you for the suggestion. I added a point of protein-protein interactions and cited two original articles in page 4. They are references 10 and 11.

2) *For reviews for mechanisms of GPI-APs sorting in polarized epithelial cells (pag. 4), I suggest to cite more recent ones that include the current advances in the field:*

Zurzolo and Simons. Glycosylphosphatidylinositol-anchored proteins. Membrane organization and transport. BBA 2016;

Lebreton, Paladino, Zurzolo. Clustering in the Golgi apparatus governs sorting and function of GPI-APs in polarized epithelial cells, Febs Letters 2019

Thank you for the suggestion. I cited two review articles in page 4-5. They are references 21 and 22.

3) *At pag. 14: the author wrote “the presence of LacCer is required for Gal transfer to the β GalNAc side chain of the GPI”, this sentence is too reductive and it is not clear for the reader how the LacCer is critical for the remodeling of GPI-anchor. Moreover, these data imply an intriguing relationship between GPI-AP biosynthesis/remodeling and glycosphingolipid biosynthesis. It would be important to debate this new concept.*

I discussed this point in page 14.

Minor comment:

For clarity, spell “CSF” (page 22, line 466)

It is now spelled as “cerebrospinal fluid”.

Referee: 2

Comments to the Author(s)

This is a comprehensive and authoritative review, that any student of modern cell biology with an interest in physiology must read.

This is a comprehensive and authoritative review, that any student of modern cell biology with an interest in cell and organismal physiology must read. The biosynthesis parts have been written with deep knowledge and simplicity that make this complex process accessible to the lay reader. The authors have highlighted many peculiar aspects of clinical phenotypes that occur due to defects in GPI-anchor biosynthesis. I have no reservation whatsoever about the outstanding quality of this review and it must be published soon.

Some suggestions for improvement:

1) A criticism if any, is the lack of mechanistic insight in the generation of the clinical phenotypes that occur due to the mutations in most places. It would be very valuable to have the perspective of one the few people who study this process (biosynthesis of GPI), and is also a leading clinician with an abiding interest in disease.

i) For example: on pg. 16 lines 337- 342 that authors state- ‘ Mammalian ARVI cDNA complemented ARVI-defective yeast, which shows the functional role of mammalian ARVI (101, 102). Recently, patients with biallelic loss-of-function mutations in ARVI have been reported (103, 104). The affected children had similar symptoms to those with inherited GPI deficiencies, such as developmental delay and seizures, which is consistent with mammalian ARVI having a putative role in GPI biosynthesis.’ However neither here nor anywhere else do we understand how GPI-biosynthesis relates to developmental delays and seizures.

I discussed roles of GPI-APs in development and seizure protection in pages 26-27 (lines 560-573).

ii) Or- on page 23: The authors blandly state ‘GPI-PLD knockout ameliorated glucose intolerance and hepatic steatosis under a high-fat and high-sucrose diet through a reduction of diacylglycerol and a subsequent decrease of PKCε activity (130).’

I explained the mechanistic relations in page 22 (lines 458-460).

iii) or on page 26: A mystery that has persisted is why PIGA mutations only cause PNH defects when they occur in stem cells, and that the somatic PIGA mutation alone does not cause PNH. ‘Indeed, similar PIGA somatic mutations are found in blood granulocytes from healthy individuals (135). When the mutant hematopoietic stem cell exhibits clonal expansion and generates large numbers of GPI-AP-defective blood cells, clinical PNH is manifested’??

I added description on this point in page 24 (lines 499-501).

iv) The authors state: line 596 Therefore, although free GPI is normal component of certain cells, abnormally accumulated free GPI is pathogenic.’ This has also been repeated in an earlier section lines 310-313. Again, this is a very intriguing result and could do with some mechanistic insight from the author.

I added discussion in page 29 (lines 625-628).

In all these cases perhaps it’s too premature for the author to speculate, but some insights about mechanism from the deep experience of the author would be valuable to pen down.

2) In Table 2: it would be useful to have a table of diseases or phenotypes associated with the mutations- or have a separate table that could delineate this.

As suggested by the reviewer, I added Table 3 summarizing diseases caused by mutations in genes in GPI pathway.

3) In Figures 2) and 3) it would be useful to map the names of the genes outlined in Table 2 to the steps indicated as 1,2,3 and so on to PIGA, M etc, so that there is a ready visual reckoning of the different genes to the steps that are involved in GPI biosynthesis.

Gene names are added to Figures 2 and 3 as suggested by the reviewer.

4) Recently, a role for PGAP2 and 3 PIGX and PIGM, respectively, in integrin signalling has been identified. This study could provide a functional insight into how the cell may use the remodelled fatty acids on GPI anchored proteins to make functional membrane microdomains (see Kalappurakkal et al. Cell. 2019 Jun 13;177(7):1738-1756). This is an interesting insight into how PIGX and PIGM deletions may affect organismal physiology, and may be discussed here.

Thank you for the suggestion. I discussed this point in page 28 (lines 589-594).

Minor:

Line 344: ..Pga1p should read PGA1p.

I fixed the typo. I follow a guideline of the Saccharomyces gene database. The product of gene PGA1 is described as Pga1p.

Line 544 'embryonic lethality, as demonstrated by knockout of the Piga gene (163).' It should read...PIGA.

I follow a guideline of the International Committee on Standardized Genetic Nomenclature for Mice. The mouse gene orthologous to human PIGA is described as Piga.